# Similarity of stream width distributions across headwater systems

George H. Allen [1], Tamlin M. Pavelsky [1], Eric A. Barefoot [1,5], Michael P. Lamb[2], David Butman [3], Arik Tashie[1] & Colin J. Gleason[4]

The morphology and abundance of streams control the rates of hydraulic and biogeochemical exchange between streams, groundwater, and the atmosphere. In large river systems, the relationship between river width and abundance is fractal, such that narrow rivers are proportionally more common than wider rivers. However, in headwater systems, where many biogeochemical reactions are most rapid, the relationship between stream width and abundance is unknown. To constrain this uncertainty, we surveyed stream hydromorphology (wetted width and length) in several headwater stream networks across North America and New Zealand. Here, we find a strikingly consistent lognormal statistical distribution of stream width, including a characteristic most abundant stream width of $32 \pm 7$ cm independent of discharge or physiographic conditions. We propose a hydromorphic model that can be used to more accurately estimate the hydromorphology of streams, with significant impact on the understanding of the hydraulic, ecological, and biogeochemical functions of stream networks.

[1] Department of Geological Sciences, The University of North Carolina at Chapel Hill, Chapel Hill, 27599 NC, USA. [2] Division of Geological and Planetary Sciences, California Institute of Technology, Pasadena, 91125 CA, USA. [3] School of Environmental & Forest Sciences and the Department of Civil & Environmental Engineering, University of Washington, Seattle, 98195 WA, USA. [4] Department of Civil and Environmental Engineering, University of Massachusetts Amherst, Amherst, 01003 MA, USA. [5] Present address: Department of Earth Science, Rice University, Houston, 77005 TX, USA. Correspondence and requests for materials should be addressed to G.H.A. (email: georgehenryallen@gmail.com)

Headwater streams[1] comprise an estimated 89% of the global fluvial network length[2,3] and are the source of water, sediment, nutrients, and organic matter for downstream systems[4]. They exhibit highly variable physical, chemical, and biotic attributes; as a result, they contribute to significant biodiversity within watersheds[5]. They are also more hydraulically coupled to hillslope and groundwater processes compared to larger streams and thus are hotspots for biogeochemical activity[2,4,6–11]. High rates of hyporheic exchange expose transported solutes to unique biogeochemical environments, with subsequent impacts on whole stream metabolism[12], nutrient cycling[13], and contaminant uptake and export[14]. Small streams are also a significant source of greenhouse gas to the atmosphere[10]. In fact, over half of the greenhouse gas emitted from the fluvial network originates from small headwater streams[2,6] (defined as Strahler stream order[15] 1–3[1]). This biogeochemical activity is, in part, a function of stream surface water geometry.

Stream width, defined here as the wetted width of flowing water within a channel, reflects natural heterogeneities along a stream such as channel margins, eddies behind large woody debris, and hyporheic exchange flow paths[9,16,17]. These heterogeneities are important because they serve as micro-environmental patches that impact temporary solute storage, material erosion and deposition, biological and ecosystem processes, and ultimately large-scale biodiversity[1,18,19]. Where it has been studied, planform stream hydromorphology is often quantified by measuring wetted width at uniformly spaced intervals along stream centerlines[19–23]. Stream width data are used in a broad array of applications including studies of hyporheic flow[16,20], open-channel hydraulics[23], material transport and erosion[22], lotic habitat[19], and stream–atmosphere gas exchange rates[2,6–8]. Stream width is also a core variable in the River Continuum Concept, an important conceptual framework that relates lotic ecosystems to stream size[1].

Despite their wide-ranging importance, there has been no published characterization of the distribution of stream widths within an entire headwater catchment. Instead, static, topographically derived flowlines are typically used to represent stream networks and infer their geometry[2,6,7,20]. However, headwater stream networks, also known as active drainage networks (ADNs), typically expand and contract with changing streamflow conditions, causing temporal fluctuations in catchment drainage density and stream surface area[24]. Temporal change in drainage density has been studied[24–27], but the simultaneous spatial dynamics of headwater stream widths remains undocumented. Instead, studies requiring stream geometry in headwater catchments usually estimate stream width distributions using hydraulic scaling principles developed for larger river systems[2,6,7,21,28]. These scaling principles produce a Pareto distribution of stream width that may be inappropriate for smaller stream networks[4,29].

In this study, we conduct field surveys to show that the distribution of stream width in headwater catchments is similar across a wide range of streamflow and physiographic conditions. We propose a stream width model that takes into account the effects of streamflow, hydraulic resistance[30], and the natural variability of channel geometry[31,32]. This model supports a new conceptual framework showing that, as ADNs expand and contract within the geomorphic channel network in response to changes in streamflow[24,25], the distribution of stream width remains approximately static. This framework can be used to accurately estimate stream surface area of ADNs if the total length of the stream network is known, with implications for stream–atmosphere biogeochemical exchange estimates.

## Results

**Field measurements.** To characterize stream width distributions in a range of headwater systems, we conducted the most comprehensive field survey to date of stream hydromorphology (wetted width and length) within seven headwater catchments (Fig. 1). The study catchments spanned a wide range of sizes, environments, geomorphologies, and streamflow conditions (see Supplementary Fig. 1 for field photos and Supplementary Table 1 for site-specific attributes). In each study catchment, we measured wetted stream width every 5 m along all flowing streams in the drainage network. Additionally, in a 48-hectare subcatchment of Stony Creek Research Watershed in North Carolina, we repeatedly surveyed stream width over a range of hydrologic conditions (Fig. 2 and Supplementary Table 2). We then analyzed the statistical distribution of stream widths from all surveys to evaluate the patterns and controls of headwater stream hydromorphology.

The stream widths of all surveys are well characterized by lognormal distributions and exhibit a mode of 32 ± 7 cm (all confidence intervals 1σ, Fig. 3). The mode width, determined

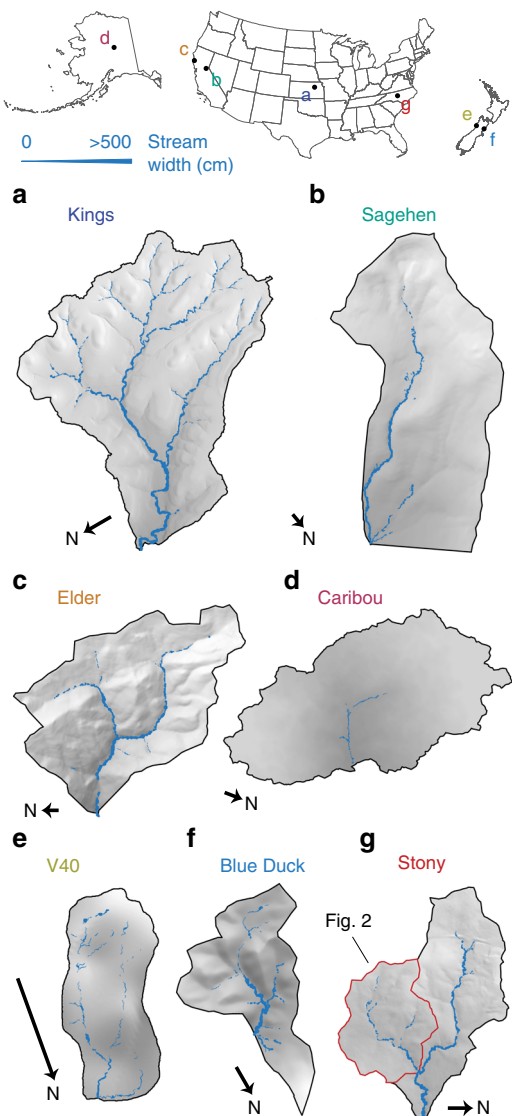

**Fig. 1** Stream width maps in study catchments. **a** K1B tributary, North Branch of Kings Creek, KS; **b** Sagehen Creek Subcatchment, CA; **c** Upper Elder Creek, CA; **d** C1 tributary of Caribou Creek, AK; **e** V40 Stream Subcatchment, NZ; **f** Blue Duck Creek Subcatchment, NZ; **g** Stony Creek Research Watershed, NC. Lengths of north arrows represent 200 m

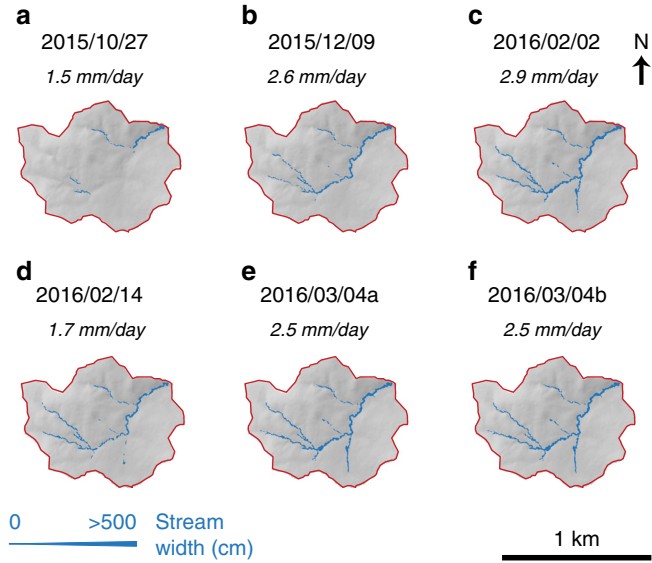

**a** 2015/10/27 *1.5 mm/day*
**b** 2015/12/09 *2.6 mm/day*
**c** 2016/02/02 *2.9 mm/day*   N ↑
**d** 2016/02/14 *1.7 mm/day*
**e** 2016/03/04a *2.5 mm/day*
**f** 2016/03/04b *2.5 mm/day*

0 >500 Stream width (cm)

1 km

**Fig. 2** Stony subcatchment repeat stream width maps with the basin-averaged runoff values in italic (dates in YYYY/MM/DD format)

using a Gaussian density kernel, is strikingly similar across all basins and does not significantly correlate with hydrologic conditions ($R^2 = 0.15$, $p = 0.39$), basin relief ($R^2 = 0.23$, $p = 0.28$), catchment area ($R^2 = 0.04$, $p = 0.69$), or drainage density ($R^2 = 0.19$, $p = 0.33$). Gamma and Weibull distributions also effectively describe the spread of stream width data (Supplementary Fig. 2 and Supplementary Table 3). The median first-order stream width is $32 \pm 8$ cm. No instances of overbank flooding were observed during the surveys but disconnections in the ADN were common, particularly in first-order streams.

**Stream width model**. To understand the origin of the characteristic lognormal distribution, we modeled stream widths by combining principles of conservation of mass, hydraulic resistance (i.e., bed roughness, skin resistance, and form drag)[30], downstream hydraulic geometry[33], and the natural variability of channel cross-sectional geometry[31,32] (Fig. 4, see Stream width model in "Methods"). Caribou and V40 catchments were excluded from the analysis because their available digital elevation models (DEMs) were of insufficient quality to produce accurate stream networks at the scales observed. The model produces stream widths that are spatially realistic (Fig. 4c) and are distributed comparably to the observations (Fig. 4d–h). The model, which is applied along the observed stream network, indicates that stream widths are primarily set by discharge and random variability in the channel geometry, from V- to U-shaped, represented by $r$ in Fig. 4b.

**Discussion**

The differences between the model outputs and the observations likely stem from simplifying assumptions regarding runoff yield, bankfull channel widths, hyporheic zone transmissivity[24], and hydraulic resistance[30], none of which were measured in the field. Perhaps the least well constrained of these factors is the variability in hyporheic zone transmissivity, a fundamental controlling property of stream hydromorphology and stream generation[24,27]. As an example, in the Elder Creek catchment, a coarse-grained poorly sorted valley-bottom sedimentary prism may be transmitting substantial hyporheic flow down valley resulting in an unexpectedly low drainage density and small, often discontinuous

streams. In contrast, Kings Creek mainly contains bedrock channels and a more fine-grained valley-bottom sedimentary prism[34]. This variability in subsurface flow is unaccounted for in the model, and may contribute to the better model fits in some catchments (e.g., Kings) than in others (e.g., Elder). In the future, a better understanding of underlying groundwater processes will improve our ability to predict headwater stream generation and hydromorphology[24].

The measured distributions of stream widths are remarkably similar given the wide variety of hydrologic conditions present during the various surveys (Fig. 3i). This insensitivity of stream width distributions to changes in catchment runoff stems from the counteracting processes of lateral stream widening and longitudinal ADN expansion. With increasing discharge, streams will simultaneously widen in place and extend upstream such that individual stream segments will increase in stream order as tributary channels are reactivated[20]. As a result, the proportional abundance of narrow streams remains roughly constant (Fig. 5). Thus, if the cumulative length ($L$) of an ADN is known, the total stream surface area (SA) may be approximated using the mean lognormal fit in Fig. 3i,

$$\mathrm{SA} = \sum \frac{L}{N} e^{\mu + \sigma X}, \qquad (1)$$

where $N$ is the number of observations, $X$ is a standard normal variable of length $N$, $\mu = \ln(32 \text{ cm})$ and $\sigma = \ln(2.3)$. We anticipate that this model may break down with the initiation of overbank flooding or when the channel network is completely occupied by the ADN.

These results hold significant implications for understanding hydrological, ecological, and biogeochemical processes occurring in headwater streams. For example, previous evaluations[2,6,7] of greenhouse gas emissions from rivers and streams estimated stream surface area using Pareto scaling laws on static USGS and international DEM-derived flowlines, which significantly underestimate the abundance of headwater streams[35]. These studies assume that median first-order stream width ranges from $160 \pm 110$ cm to $315$ cm[2,3,6,7], an order of magnitude greater than observed in this study. To evaluate the impact of these differences, we compare our observations against existing flowline data sets currently used in biogeochemical studies to calculate stream microbial enzyme activity, nutrient uptake, and nutrient limitation[11], and surface emissions of $CO_2$[2,6,7] (see carbon efflux estimates in "Methods"). We find that the dynamic expansion and contraction of ADNs causes significant temporal variability in greenhouse gas emissions in headwater stream networks. In the repeat stream width surveys, estimated $CO_2$ efflux quadruples in response to a doubling in runoff (Supplementary Table 4). Among the physiographically contrasting catchments, we find that $CO_2$ efflux calculations based off of the USGS flowline data sets differ from our survey-based estimates by as much as 100% (RMSE = 17.7 Mg-C Yr$^{-1}$). Using Eq. (1) to estimate surface area yields, $CO_2$ efflux values that are more similar to our survey-based efflux estimates than the USGS flowline-based efflux estimates (RMSE = 4.17 Mg-C Yr$^{-1}$). The differences between $CO_2$ efflux estimates arise from the dissimilarity of stream network length and width distributions between our observations, Eq. (1), and the USGS flowline data sets.

In conclusion, our observations suggest that stream widths in headwater networks are lognormally distributed, rather than Pareto distributed, and that the most common stream width is substantially narrower than previously assumed. This lognormal distribution can be used to more accurately estimate stream surface area in small headwater catchments if the total length of the stream network is given, with implications for

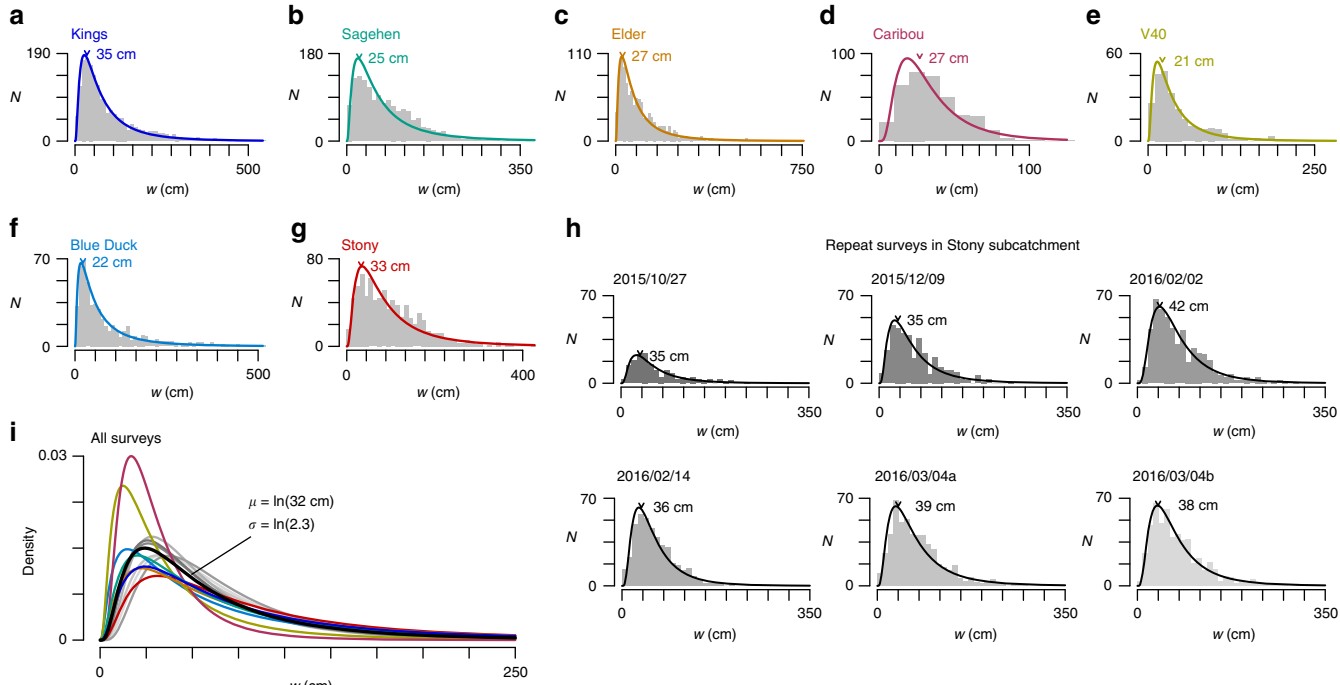

**Fig. 3** Headwater stream width distributions. **a–h** Stream width histograms with lognormal fits for each survey. Length values indicate the mode stream width in each catchment. **i** Composite probability density functions of lognormal fits with mean fit in black. *μ* and *σ* are the lognormal location and scale parameters in Eq. (1). *N*: measurement frequency, *w*: stream width

stream–atmosphere gas exchange estimates. We find that the dynamic nature of ADNs causes significant variability in greenhouse gas emissions in headwater stream networks. Significant work remains to understand how stream width and network length is controlled by groundwater interactions. Our limited data set of 13 surveys likely does not describe the full range of width distributions in headwater stream networks, especially in arid and humid tropical environments. This study's observation of a characteristic most abundant stream width suggests the existence of a most abundant stream depth and stream velocity, an unexplored hydrologic framework that may yield greater knowledge of stream generation processes and habitat distributions within stream networks.

## Methods

**Field measurements.** In each of the seven physiographically contrasting study catchments, we paced along all streams within the stream network and measured wetted stream width every 5 m[36]. For the repeat surveys in the Stony subcatchment, we flagged streams every 5 m and surveyed wetted stream widths at each flag over a range of hydrologic conditions. For the repeat surveys in Stony subcatchment, we only analyzed surveys that were collected while streamflow was below the 90th percentile of the streamflow record in order to remove the potential influence of overbank flooding (Supplementary Table 2).

We defined a stream as flowing water within a channel[26], including transient channels formed in leaf litter and debris. We measured wetted stream width with a standard tape measure or, where a tape measure was not practical, with a laser range finder. In multichannel streams, we added the stream widths from all channels together or we visually approximated the percentage of the total width that was dry. We quantified measurement error by repeatedly surveying stream width along a 175-m long stream segment located at the lowermost segment of the Stony subcatchment. We surveyed the segment five times within 1.5 h and then compared the width measurements to calculate standard error of 3 cm.

In each catchment, we collected between 160 and 1797 (mean *N* = 672) stream width measurements. We mapped ADNs with a continuous tracking GPS device, or where necessary, by hand on a topographic map or on optical remotely sensed imagery. We removed 36 survey points (2.5%) from our analysis of Sagehen Creek, where snow completely or partially obscured the stream surface. We approximated relative hydrological conditions in each physiographically contrasting study catchment by calculating the streamflow percentile and catchment-averaged runoff during the day(s) we surveyed streams relative to the entire gage record (Supplementary Table 1). Stream gages were often located nearby or downstream

from the study catchments and thus the runoff and discharge percentiles presented in Supplementary Table 1 indicate a general characterization of catchment wetness.

**Statistics.** We described the statistical distribution of stream widths within each study basin by fitting lognormal, gamma, Weibull, and Pareto distributions to the stream width data using maximum likelihood estimation. We quantified model goodness of fit using the Pearson's chi-square goodness of fit test[37] and the non-parametric two-sided one sample Kolmogorov–Smirnov goodness of fit test[37] (Supplementary Fig. 2 and Supplementary Table 3). Using a Gaussian density kernel with a bandwidth of 10 cm, we calculated the mode stream width in each basin. Kernel bandwidth was determined using the normal reference distribution optimal bandwidth selection technique[37]. We correlated the mode width to physical conditions (hydrologic conditions, basin relief, catchment area, and drainage density) between physiographically contrasting catchments using the correlation of determination ($R^2$) and *p* value with significance level, *α* = 0.05. We calculated total stream surface area by summing the product of each stream width and length measurement within a catchment.

**Stream width model.** We modeled stream width in each catchment using relationships between stream channel shape, hydraulic resistance, drainage area, and discharge. While this model shares some conceptual similarities to preceding studies[31,32,38,39], it is, to our knowledge, a novel synthesis of downstream hydraulic geometry, at-a-station hydraulic geometry, and natural variability in stream channel cross-sectional geometry. Our model begins with the analytical relationship for at-a-station hydraulic geometry presented by Dingman[31]. The cross-sectional shape of stream channels has been modeled as a variety of geometries including triangular, parabolic, trapezoidal, and rectangular[40]. Here, we simulated these channel shapes by varying a single shape parameter, *r*, such that for any wetted depth less than bankfull depth,

$$h = h_{\text{bf}} \left( \frac{w}{w_{\text{bf}}} \right)^r, \quad (2)$$

where $h_{\text{bf}}$ is bankfull channel depth, *w* is wetted stream width, and $w_{\text{bf}}$ is bankfull channel width (Fig. 4a)[31,32]. Setting *r* = 1 yields a triangular cross section, and increasing its value beyond 1 yields an increasingly concave (or flat-bottomed) parabolic channel shape (Fig. 4b).

Within a channel, the law of conservation of mass relates stream discharge to channel shape,

$$Q = uA_{\text{s}} \quad (3)$$

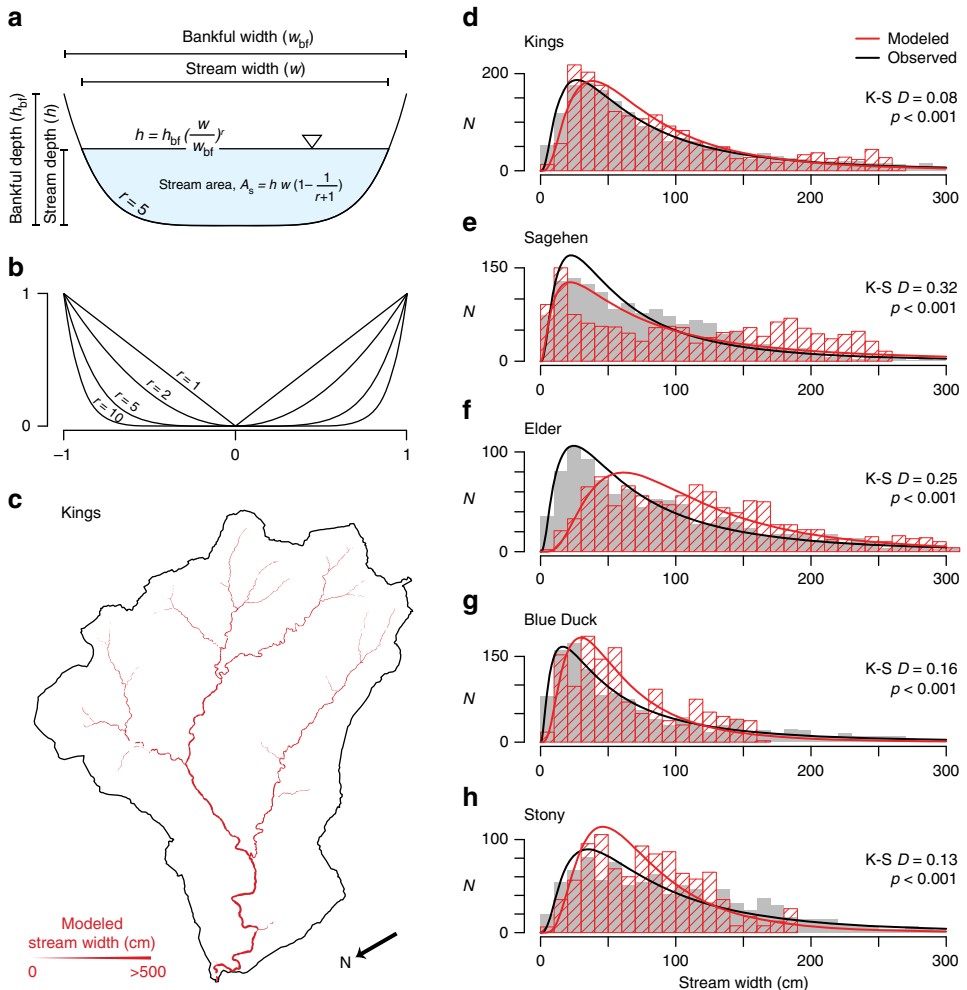

**Fig. 4** Origin of the lognormal stream width distribution. **a** Schematic channel cross section; **b** schematic relationship between the shape parameter, *r*, and channel cross-sectional shape; **c** map of modeled stream widths in example study catchment. Length of north arrow is 200 m. **d**–**h** The distribution of modeled stream widths compared to field observations. *N*: measurement frequency. K–S *D*: Kolmogorov–Smirnov test statistic. *p*: calculated probability

where *u* is mean streamflow velocity and $A_s$ is the cross-sectional area of the stream and is calculated as,

$$A_s = hw\left(1 - \frac{1}{r+1}\right). \qquad (4)$$

We modeled flow velocity using a form of the Manning–Strickler relation for flow resistance[30],

$$u = 8.1(ghS)^{\frac{1}{2}}\left(\frac{h}{k}\right)^{\frac{1}{6}}, \qquad (5)$$

where *g* is gravitational acceleration, *S* is channel bed slope, and *k* is a bed roughness length scale equivalent to,

$$k = \left(8.1g^{\frac{1}{2}}n\right)^6, \qquad (6)$$

where *n* is the Gauckler–Manning friction coefficient[41]. Combining Eq. (2–5) yields a relationship, similar to a generalized relationship presented in Dingman[31], between stream width and other hydraulic and geomorphic parameters:

$$w = Q^{\frac{3}{5r+3}}w_{bf}^{\frac{r-1}{r+3/5}}\left(8.1(gS)^{\frac{1}{2}}k^{-\frac{1}{6}}\left(\frac{w_{bf}}{h_{bf}}\right)^{-\frac{3}{5}}\left(1-\frac{1}{r+1}\right)\right)^{-\frac{3}{5r+3}}. \qquad (7)$$

Bankfull channel width scales with upstream drainage area (*A*) as a power law,

$$w_{bf} = \alpha A^\beta, \qquad (8)$$

where $\alpha$ and $\beta$ are empirical constants[42–45]. We calculated the values of $\alpha$ and $\beta$ to be 0.008 and 0.42, respectively, using least squares regression on a global compilation of 307 paired log-transformed $w_{bf}$ and *A* measurements[45] ($R^2 = 0.68$, $p < 0.001$), values similar to those found by previous work[42–44]. Note that using the empirical $\beta$ value of 0.42, rather than the dimensionally consistent $\beta$ value of 0.5, propagates a minor imbalance in the dimensions of Eq. (7). We used the same global database to find that the median bankfull width to depth ratio $h_{bf}/w_{bf}$ is 14 in Eq. (7), for streams with upstream drainage areas within the range we surveyed in this study. Previous studies have reported $h_{bf}/w_{bf}$ ratios varying from 1.5 to 35, and the value we used falls within this range[46–49].

At each stream width observation site, *i*, we calculated stream width by combining Eqs. (7) and (8),

$$w_i = Q_i^{\frac{3}{5r_i+3}}\left(\alpha A_i^\beta\right)^{\frac{r_i-1}{r_i+3/5}}\left(8.1(gS_i)^{\frac{1}{2}}k^{-\frac{1}{6}}\left(\frac{w_{bf}}{h_{bf}}\right)^{-\frac{5}{3}}\left(1-\frac{1}{r_i+1}\right)\right)^{-\frac{3}{5r_i+3}}. \qquad (9)$$

We computed $A_i$ and $S_i$ from DEMs listed in Supplementary Table 1[50] and calculated slope over the length scale of the DEM resolution used. To estimate $Q_i$ at each survey location, we used conservation of mass within a drainage basin,

$$Q_i = Q_g\frac{A_i^c}{A_g^c}, \qquad (10)$$

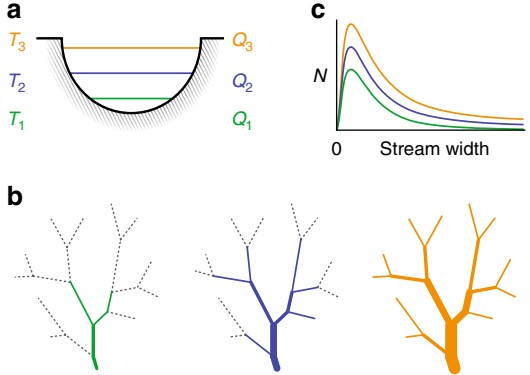

**Fig. 5** Conceptual model of relationship between changing streamflow conditions and the stream width distributions. As discharge increases ($Q_1 < Q_2 < Q_3$): **a** wetted width increases at a representative channel cross section; **b** the ADN extends into narrower lower-order channels, increasing the total length of narrow streams (dotted lines represent channels without flow); and **c** the phenomena described by **a** and **b** manifest in the similar width distributions independent of streamflow conditions

where $Q_g$ and $A_g$ are the discharge and drainage area at the stream gage (Supplementary Table 1) and set $c = 1$[51]. Setting $c$ to unity is a common practice in hydrology[52], although it is a relationship developed in humid environments. We set the Gauckler–Manning friction coefficient, $n = 0.04$ ms$^{-1/3}$, as commonly assumed for mountain streams with gravel/cobble beds[22,40]. To represent natural diversity in stream channel shape, we randomly varied the value of $r_i$ in Eq. (9) between 1 and 10, and thus captured channel geometries ranging from a triangle to a shape approaching a rectangle (Fig. 4b). In a separate analysis, we drew values of $r_i$ from a normal distribution ($\mu = 5$, $\sigma = 2$), rather than a uniform distribution, which yielded similar results. We characterized the statistical difference between the modeled and the surveyed stream widths using the two-sample Kolmogorov–Smirnov test[37] (Fig. 4 and Supplementary Table 3).

Classic hydraulic geometry relationships are reflected in the exponents of Eq. (9). The first factor in Eq. (9), $Q_i$, represents the at-a-station hydraulic geometry[33] component of the model, the second, $aA_i^\beta$, represents the downstream changes in channel width, rather than wetted stream width, and the remaining factors encapsulate the influences of flow resistance and channel geometry. If $r = 2$, then $w_i \propto Q_i^{\frac{3}{13}} = Q_i^{0.23}$, which is similar to classic power-law relations for at-a-station hydraulic geometry[33], where $w = Q^{0.26}$. Similarly, for classic downstream hydraulic geometry[33], where $w = Q^{0.5}$, if $r = 2$, then $w \propto Q_i^{0.23} A_i^{\beta 5/13}$, and if $A_i \propto Q_i$ and $\beta = 0.42$, then $w_i \propto Q_i^{0.39}$. Equation (9) predicts that stream width positively relates to bed roughness because a larger flow cross-sectional area is required to transmit the same amount of discharge in a rougher channel. However, the modeled widths produced from Eq. (9) are not highly sensitive to the Gauckler–Manning friction parameter, $n$, and produce reasonable width distributions over a range of roughness coefficients ($0.03 < n < 0.1$) common in the modeled stream networks[40,53].

At very low flows, where the flow depth approaches the scale of the bedload, the Manning–Strickler relationship for open-channel flow, shown in Eq. (5), breaks down[54]. Ferguson[55] proposed a hydraulic resistance equation that may be applied at shallow flows,

$$\left(\frac{8}{f}\right)^{1/2} = \frac{a_1 a_2 (h/D_{84})}{\left[a_1^2 + a_2^2 (h/D_{84})^{5/3}\right]^{1/2}} \tag{11}$$

where $f$ is the Darcy–Weisbach friction factor, $a_1$ and $a_2$ are coefficients that range from 7–8 and 1–4 respectively, and $D_{84}$ is the 84th percentile bedload grain diameter. Equation (11) predicts that with decreasing $h/D_{84}$, hydraulic resistance increases, which would theoretically increase the hydraulic radius[56,57]. Because we do not have measurements of $D_{84}$ this widening effect is not implemented in our model but it could be included in future work to improve representation of headwater stream hydromorphology.

**Carbon efflux estimates**. To estimate carbon efflux within each North American catchment, we used topographically derived USGS flowlines to estimate stream surface area and stream gas transfer velocity, and $CO_2$ efflux methods described in detail by Butman et al.[6]. We then compared these efflux estimates to estimates based off of our field observations. We did not conduct this analysis in the New Zealand watersheds because no suitable flowline data set was available in this region. In the conterminous United States, we used NHDPlus V2 flowlines (http://www.horizon-systems.com/nhdplus/NHDPlusV2_home.php) to calculate carbon

efflux, and in the Caribou Creek catchment in Alaska, where NHDPlus data are unavailable, we used EDNA flowlines (http://edna.usgs.gov). NHDPlus V2 and EDNA flowlines are derived from merging the Nation Hydrology Dataset (NHD) with the National Elevation Dataset (NED). The NHD contains perennial, intermittent, and ephemeral streamlines that were field mapped by the USGS. Thus, the exact conditions in which the NHDPlus V2 flowlines represent are poorly constrained.

In each study catchment, we used the median (5th and 95th percentile ranges) dissolved $CO_2$ concentrations and temperatures for first-order stream systems in the larger two-digit USGS hydrologic unit code (HUC) region. For each HUC, we used established hydraulic geometry equations[3] to estimate first-order stream velocity from the stream slope and the median discharge values provided by the USGS flowline data sets. Using these stream velocities, we estimated the median $CO_2$ gas transfer velocity for first-order streams in each HUC after Raymond et al.[2]. For the Stony Creek Research Watershed where no NHDPlus flowlines exist, we used the discharge measurements taken at the catchment outlet to directly estimate velocity. We calculated the amount of stream surface area using three different methods: (1) field measured surface area; (2) Eq. (1) derived surface area; and (3) USGS flowline-derived surface area[2]. Then, we ran a Monte Carlo simulation to estimate the median and 5th–95th percentile ranges of potential $CO_2$ efflux (Supplementary Table 4).

**Data availability**. All the codes used in the analysis and production of figures in this paper can be obtained at https://github.com/geoallen/streamWidthAnalysis2017/. The stream width measurements, from which the lognormal distributions are derived, are available on a Zenodo digital repository (doi:10.5281/zenodo.1034384).

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

## Acknowledgements

Financial support for this research was provided by NASA New Investigator Program grant #NNX12AQ77G, a UNC Geological Sciences Graduate Student Fellowship, and a Geological Society of America Graduate Student Research Grant to George Allen. Emily Beckham and Elizabeth Henry assisted with fieldwork. Dr. Margaret Zimmer and Dr. Brian McGlynn helped facilitate fieldwork and provided stream gage data in the Stony Creek Research Watershed. Dr. Xiao Yang assisted with theoretical derivations. Dr. Jeremy Jones provided Caribou Creek stream gage data.

## Author contributions

G.H.A. and T.M.P. conceived the research idea. G.H.A. designed and performed the analysis, drafted the figures, and wrote the manuscript with input from all coauthors. G.H.A. and E.A.B. conducted the fieldwork, with help from A.T. and D.B. M.P.L. and C.J.G. helped with theory and D.B. calculated carbon fluxes.

## Additional information

**Competing interests:** The authors declare no competing financial interests.

