## [Peer Review File · Nature Communications]

Reviewers' comments:

Reviewer #1 (Remarks to the Author):

I have now read this manuscript for the fourth time. Each time new issues have cropped up, either because they are really new or because I didn't see them before. In the present case there remain two technical issues.

The first is a technical problem with equation 1: Sigma is given as $\ln(2.3 \text{ cm})$, but this is dimensionally incorrect. One can see the problem directly; if one changes the units of cm to m, for example, then $\ln(0.023 \text{ m})$ becomes much larger, and negative. Sigma expresses the width of the distribution in natural log units (that is, it is a dimensionless multiplier). The usual formula for sigma, in terms of the (non-log-transformed) mean and variance of the distribution, is $\sigma^2 = (1 + \text{var}(x)) / (\text{mean}(x)^2)$. As one can see, sigma is and must be dimensionless. (Indeed, mu has to be dimensionless too, as must any argument of a logarithm or exponential – otherwise the dimensions of the result don't make sense.)

Putting the problem differently, if mu is $\ln(32 \text{ cm})$ and sigma is $\ln(2.3 \text{ cm})$, then when one evaluates equation (1), one gets a result with dimensions of AREA (that is, cm^2), not width (because the addition of mu and sigma in the exponential implies multiplication).

The second problem concerns equation 10. Although it seems intuitively reasonable to make discharge proportional to drainage area – and this is arguably correct for channelized plus subsurface flow – it cannot be right as a guide to the channel width of headwater streams. The problem is that equation 10 implies that there will always be surface flow, and thus there will always be a channel, even at tiny drainage areas, and thus it implies that channels will go all the way to the divide – and, more problematically, that they will go to the divide EVERYWHERE, thus filling the entire land surface with channels. That's not the way things work in the real world. In the real world, there is a minimum drainage area required to support a channel (see the Montgomery and Dietrich papers from about 30 years ago...)

Combining equation 10 with equation 9 gives the result that there should be channels of finite width even down to tiny drainage areas (for $r=2$ and $\beta=0.42$, equation 9 predicts that width scales as A with an exponent of almost exactly 0.4). This may be empirically reasonable for drainage areas at which channels actually exist, but it also predicts that channels should occupy the entire landscape, and they don't. Given that the whole point of the model is to explain the distribution of channel widths, it's surprising that this issue apparently hasn't been noticed.

Putting the problem differently: in the real world, channel widths go to zero at channel heads, and these channel heads typically have substantial drainage areas. This behavior is incompatible with the channel width model as stated.

Finally, there are STILL incoherent bits of language in the manuscript, such as "presented in by Dingman" on line 320 or "between the modeled and surveyed streams widths" on line 369. Particularly given that this problem has been pointed out repeatedly (and the authors are native English speakers), is it too much to ask for careful proofreading of a manuscript submitted for publication at this level?

The issues raised here are not fundamental, and I assume they can be dealt with in short order. As I have consistently maintained in my previous reviews, this work clearly deserves to be widely read. I continue to support its publication.

Reviewer #2 (Remarks to the Author):

I believe that Allen et al. have presented an intriguing field data set and compelling statistical analysis, and they have highlighted some of the important implications of this work for understanding carbon fluxes from headwater streams. Their attention to previous reviewer comments has improved this manuscript, and I have only a couple of lingering/new concerns about this version.

(1) More attention is needed in the presentation of the K-S results of the model (Fig. 3). It seems more appropriate to clearly point to differences between the visual and statistical results, as well as the implication for future opportunities for tackling this problem (and the stream width prediction problem, more broadly).

(2) I think that readers would benefit from more of your insight into the effects of some of your modeling choices that you've summarized in the response to reviewers. Recognizing that space is limited, some summary points would nonetheless be helpful (e.g., perhaps this could allow those who might be interested in exploring this further what range of n or d/D you've explored, and under which conditions w estimates appear unreasonable with this model).

Detailed notes below

24/ footnoted, parenthetical stream order in 1st sentence might put off broader audience. Can this definition come later?

34/ perhaps add "as we show here"?

82/ don't the K-S D values show significant differences in distributions for all sites at $p < 0.001$ level? Visually, the fits are not too bad, but this is awkward. When the manuscript states, that the model matches "most closely" in 4/5 catchments, how are you making this division? Do you exclude Elder? Clarify what you really mean here: it doesn't visually match well for one site? (And not that the model differs significantly from observations for one site, right?)

85/ "the significant differences" - this will need to be revised in accordance with above. i.e., clarify that this isn't just at the one site where it "least closely" matches, but that the model isn't great so far at matching the whole distribution. This seems like a strong requirement given the extensive assumptions you're having to make (lines 86-87), and you might want to explore whether the median/modal widths are significantly different.

107/ fluxes associated with dynamics vs. static wet/dry conditions. Rewetting as imp. period (citations)

118-124/ it seems like this section repeats some of the previous paragraph, and could be integrated there more efficiently (to allow for more discussion above/below)

251/ specify yyyy/mm/dd for global readership?

263/ typo: d-h

309/ specify what "physical conditions" means

320/ typo: "in by Dingman"

363-365/ It would be helpful to comment (briefly) here on the impact of this assumption in light of well-known problems with Manning's approach in shallow flow (and perhaps in recent work by Bonetti et al. 2017 Journal of Fluid Mechanics). You've addressed part of this in the previous reply to R2 (reasonable variations in n likely have little impact; model limitations include obvious overestimates of w as d/D approaches 0). I agree that the full analyses should not be included in the manuscript, but a summary here would help readers.

377/ parallel to previous sentence: is this similar to typical patterns? brief discussion of this shift in power-law scaling exponent would be helpful.

390/ "However," not "Thus," correct? What are the implications of this? Would be helpful to preview that you'll compare these to your maps here, not wait until line 400.

Note: somewhat erratic hyphen usage has emerged in some parts of this revision - please work with a copy editor before publication

Citations

Bonnetti S. et al. 2017. Manning's formula and Strickler's scaling explained by a co-spectral budget model, *J. Fluid Mechanics*, 812: 1189-1212.

Reviewer #1 (Remarks to the Author):

I have now read this manuscript for the fourth time. Each time new issues have cropped up, either because they are really new or because I didn't see them before. In the present case there remain two technical issues.

The first is a technical problem with equation 1: Sigma is given as $\ln(2.3 \text{ cm})$, but this is dimensionally incorrect. One can see the problem directly; if one changes the units of cm to m, for example, then $\ln(0.023 \text{ m})$ becomes much larger, and negative. Sigma expresses the width of the distribution in natural log units (that is, it is a dimensionless multiplier). The usual formula for sigma, in terms of the (non-log-transformed) mean and variance of the distribution, is $\sigma^2 = (1 + \text{var}(x)) / (\text{mean}(x)^2)$. As one can see, sigma is and must be dimensionless. (Indeed, mu has to be dimensionless too, as must any argument of a logarithm or exponential – otherwise the dimensions of the result don't make sense.)

Putting the problem differently, if mu is $\ln(32 \text{ cm})$ and sigma is $\ln(2.3 \text{ cm})$, then when one evaluates equation (1), one gets a result with dimensions of AREA (that is, cm^2), not width (because the addition of mu and sigma in the exponential implies multiplication).

We thank Referee #1 for noticing this technical problem in Equation 1. We removed the units from sigma and now the units are of length, not area since $\exp(\ln(\text{cm}) + \ln(k)) = k \cdot \text{cm}$, where k is a constant. This equation also produces the correct result when converting from cm to meters.

As an explanation, for a log-normal distribution, it is customary to report the geometric mean = $\exp(\mu)$, which is equivalent to the median. mu has units of $\ln(x)$, so the geometric mean has units of x , which is length in this case. sigma is the shape parameter of the log distribution, it is unitless, and thus the geometric standard deviation is $\exp(\sigma)$ and is also unitless. If Referee #1 is looking for something different, we welcome any suggested solutions.

The second problem concerns equation 10. Although it seems intuitively reasonable to make discharge proportional to drainage area – and this is arguably correct for channelized plus subsurface flow – it cannot be right as a guide to the channel width of headwater streams. The problem is that equation 10 implies that there will always be surface flow, and thus there will always be a channel, even at tiny drainage areas, and thus it implies that channels will go all the way to the divide – and, more problematically, that they will go to the divide EVERYWHERE, thus filling the entire land surface with channels. That's not the way things work in the real world. In the real world, there is a minimum drainage area required to support a channel (see the Montgomery and Dietrich papers from about 30 years ago...)

Combining equation 10 with equation 9 gives the result that there should be channels of finite width even down to tiny drainage areas (for $r=2$ and $\beta=0.42$, equation 9 predicts that width scales as A with an exponent of almost exactly 0.4). This may be empirically reasonable for drainage areas at which channels actually exist, but it also predicts that channels should occupy the entire landscape, and they don't. Given that the whole point of the model is to explain the distribution of channel widths, it's surprising that this issue apparently hasn't been noticed.

Putting the problem differently: in the real world, channel widths go to zero at channel heads, and these channel heads typically have substantial drainage areas. This behavior is incompatible with the channel width model as stated.

The stream width model presented here is only applied to locations where we know there is a stream a priori. Thus, this model does not explain where streams begin, which falls into the realm of stream generation processes. It simply explains the distribution if the stream network extent is known. To add more clarity to the main text we have added, "The model, which is applied along the observed stream network, indicates that..." (L108-109). Also, at the suggestion of Referee #2, we included a paragraph discussing remaining work to improve the stream width model. In this paragraph we explain that this model does not take into account complicated spatial variability in hyporheic zone transmissivity, which is a key control of stream generation (L112-124). Additionally, the introduction includes the statement that "Stream width [is] defined here as the wetted width of flowing water within a channel." (L46) and this is also stated in the Methods (L181). Thus, streams are, by definition, limited to the channel network, with channel heads controlled by erosional processes as Referee #1 notes above.

Finally, there are STILL incoherent bits of language in the manuscript, such as "presented in by Dingman" on line 320 or "between the modeled and surveyed streams widths" on line 369. Particularly given that this problem has been pointed out repeatedly (and the authors are native English speakers), is it too much to ask for careful proofreading of a manuscript submitted for publication at this level?

We apologize for the unnoticed mistakes in the writing. We had someone who has never read the manuscript before proof read the text before this resubmission in hopes of catching all writing errors.

The issues raised here are not fundamental, and I assume they can be dealt with in short order. As I have consistently maintained in my previous reviews, this work clearly deserves to be widely read. I continue to support its publication.

We thank Referee #1 for carefully reviewing our manuscript and providing insightful feedback throughout the review process. Their comments have vastly improved the quality of this study.

Reviewer #2 (Remarks to the Author):

I believe that Allen et al. have presented an intriguing field data set and compelling statistical analysis, and they have highlighted some of the important implications of this work for understanding carbon fluxes from headwater streams. Their attention to previous reviewer comments has improved this manuscript, and I have only a couple of lingering/new concerns about this version.

(1) More attention is needed in the presentation of the K-S results of the model (Fig. 3). It seems more appropriate to clearly point to differences between the visual and statistical results, as well as the implication for future opportunities for tackling this problem (and the stream width prediction problem, more broadly).

As suggested, we removed the statistically ambiguous language that we previously used to describe the goodness of fit between the model output and the field observations. We also added a paragraph in the Discussion and conclusions section describing, in our view, the most poorly constrained factor hampering our ability to model stream hydromorphology: variability in hyporheic zone transmissivity (L113-130).

(2) I think that readers would benefit from more of your insight into the effects of some of your modeling choices that you've summarized in the response to reviewers. Recognizing that space is limited, some summary points would nonetheless be helpful (e.g., perhaps this could allow those who might be interested in exploring this further what range of n or d/D you've explored, and under which conditions w estimates appear unreasonable with this model).

We included a paragraph in the Methods summarizing our choice of modeling parameters (flow resistance and flow depth), including the limits of these parameters in our model (L294-310).

Detailed notes below

24/ footnoted, parenthetical stream order in 1st sentence might put off broader audience. Can this definition come later?

As suggested, we moved the definition of headwater streams to near the end of the paragraph (L43).

34/ perhaps add "as we show here"?

We suggest that including this results-oriented clause in the introduction would be getting a little ahead of ourselves, so we have decided not to include it here. However, if the editors feel strongly that it should be added to the text, we are willing to add it.

82/ don't the K-S D values show significant differences in distributions for all sites at $p < 0.001$ level? Visually, the fits are not too bad, but this is awkward.

We have replaced the word "similarly", which may have statistical connotations, with the word "comparably" because we believe that we should point out that the models do produce visually-similar distributions (L108).

When the manuscript states, that the model matches "most closely" in 4/5 catchments, how are you making this division? Do you exclude Elder? Clarify what you really mean here: it doesn't visually match well for one site? (And not that the model differs significantly from observations for one site, right?)

We removed this clause from the manuscript because, as pointed out here, it was imprecise and statistically meaningless.

85/ "the significant differences" - this will need to be revised in accordance with above. i.e., clarify that this isn't just at the one site where it "least closely" matches, but that the model isn't great so far at matching the whole distribution. This seems like a strong requirement given the extensive assumptions you're having to make (lines 86-87), and you might want to explore whether the median/modal widths are significantly different.

In response to the previous comment, we removed any mention of the model run matching the observations less well in Elder than the other catchments. Thus, we do not think that readers will mistakenly believe that this sentence implies that the only significant difference is in Elder catchment.

107/ fluxes associated with dynamics vs. static wet/dry conditions. Rewetting as impt. period (citations)

As suggested, we added “stream microbial enzyme activity, nutrient uptake and nutrient limitation” and cite Hill et al., (2010), a study that uses NHDPlus to study these processes. (L144-147)

118-124/ it seems like this section repeats some of the previous paragraph, and could be integrated there more efficiently (to allow for more discussion above/below)

While we agree that this section is somewhat repetitive, this text is located at the beginning of the concluding paragraph, which summarizes and discusses the key points of the manuscript. The format does make this section similar to previous text, but we view it as a redundancy necessary for the reader to be able to efficiently obtain the main points of the study in the concluding paragraph. If the editor feels that it is important to do so, we are happy to modify the paragraph.

251/ specify yyyy/mm/dd for global readership?

We included “(dates in YYYY/MM/DD format)” as suggested.

263/ typo: d-h

We changed “d-c” to “d-h” as suggested.

309/ specify what “physical conditions” means

We added “(hydrologic conditions, basin relief, catchment area, drainage density)” to improve clarity (L207-208).

320/ typo: “in by Dingman”

We removed “in” as suggested.

363-365/ It would be helpful to comment (briefly) here on the impact of this assumption in light of well-known problems with Manning’s approach in shallow flow (and perhaps in recent work by Bonetti et al. 2017 Journal of Fluid Mechanics). You’ve addressed part of this in the previous reply to R2 (reasonable variations in n likely have little impact; model limitations include obvious overestimates of w as d/D approaches 0). I agree that the full analyses should not be included in the manuscript, but a summary here would help readers.

As suggested, we included a paragraph that summarizes our choice of model parameters (n and depth/roughness ratio) and discussed the range limits of these parameters in our model. We also commented on the problems with Manning’s formula when applied to shallow flow and included the Bonetti et al., (2017) citation (L278-292).

377/ parallel to previous sentence: is this similar to typical patterns? brief discussion of this shift in power-law scaling exponent would be helpful.

To clarify, we defined the exponential values of classic at-a-station and downstream hydraulic geometry (L273-276). Changing the value of the shape parameter, r , will change the values of the exponents, but we believe that exploring this relationship in the text is beyond the scope of this study. However, if R2 would like us to expand on this aspect of the model, we are happy to add further discussion in the Methods.

390/ “However,” not “Thus,” correct? What are the implications of this? Would be helpful to preview that you’ll compare these to your maps here, not wait until line 400.

Actually, we are trying to communicate that the extent of NHD flowlines do not correspond to any precise hydrological conditions (e.g. median flow), so “thus” is the word we intended to write. To improve readability, we added “We then compared these efflux estimates to estimates based off of our field observations” (L297-298).

Note: somewhat erratic hyphen usage has emerged in some parts of this revision - please work with a copy editor before publication

We corrected the hyphen usage errors throughout the manuscript.

Citations

Bonnetti S. et al. 2017. Manning's formula and Strickler's scaling explained by a co-spectral budget model, J. Fluid Mechanics, 812: 1189-1212.

We thank Referee #2 for their attention to detail and for their helpful comments that have significantly improved this manuscript.

REVIEWERS' COMMENTS:

Reviewer #1 (Remarks to the Author):

The authors have responded adequately to the latest round of review comments, and the manuscript has evolved greatly over the several rounds of review. It deserves to be read. You should publish it.

There are still minor glitches, such as at line 266: "between the modeled and the surveyed streams widths" rather than "stream widths". But one assumes that these will be cleaned up in the proofreading process.

Similarity of stream width distributions across headwater systems
Allen et al.

REBUTTAL TO REVIEWERS' COMMENTS

Responses to the reviewer comments are shown in blue below.

Reviewer #1 (Remarks to the Author):

The authors have responded adequately to the latest round of review comments, and the manuscript has evolved greatly over the several rounds of review. It deserves to be read. You should publish it.

We thank Referee #1 for their rigorous and useful comments throughout the review process.

There are still minor glitches, such as at line 266: "between the modeled and the surveyed streams widths" rather than "stream widths". But one assumes that these will be cleaned up in the proofreading process.

As suggested, we fixed the writing error on line 266 and have proofread the manuscript in hopes of catching all writing mistakes.